# Increase of Input Resistance of a Normal-Mode Helical Antenna (NMHA) in Human Body Application

**DOI:** 10.3390/s20040958

**Published:** 2020-02-11

**Authors:** Norsiha Zainudin, Tarik Abdul Latef, Narendra Kumar Aridas, Yoshihide Yamada, Kamilia Kamardin, Nurul Huda Abd Rahman

**Affiliations:** 1Department of Electrical Engineering, Faculty of Engineering, University of Malaya, 50603 Kuala Lumpur, Malaysia; narendra.k@um.edu.my; 2Malaysia-Japan International Institute of Technology (MJIIT), Universiti Teknologi Malaysia, 54100 Kuala Lumpur, Malaysia; yoshihide@utm.my (Y.Y.); kamilia@utm.my (K.K.); nurulhuda0304@uitm.edu.my (N.H.A.R.); 3Antenna Research Centre, Faculty of Electrical Engineering, Universiti Teknologi MARA, 40450 Shah Alam, Selangor, Malaysia

**Keywords:** normal-mode helical antenna (NMHA), input resistance, conductivity, human body

## Abstract

In recent years, the development of healthcare monitoring devices requires high performance and compact in-body sensor antennas. A normal-mode helical antenna (NMHA) is one of the most suitable candidates that meets the criteria, especially with the ability to achieve high efficiency when the antenna structure is in self-resonant mode. It was reported that when the antenna was placed in a human body, the antenna efficiency was decreased due to the increase of its input resistance (*R_in_*). However, the reason for *R_in_* increase was not clarified. In this paper, the increase of *R_in_* is ensured through experiments and the physical reasons are validated through electromagnetic simulations. In the simulation, the *R_in_* is calculated by placing the NMHA inside a human’s stomach, skin and fat. The dependency of *R_in_* to conductivity (*σ*) is significant. Through current distribution calculation, it is verified that the reason of the increase in *R_in_* is due to the decrease of antenna current. The effects of *R_in_* to bandwidth (BW) and electrical field are also numerically clarified. Furthermore, by using the fabricated human body phantom, the measured *R_in_* and bandwidth are also obtained. From the good agreement between the measured and simulated results, the condition of *R_in_* increment is clarified.

## 1. Introduction

Biotelemetry is defined as the remote detection and measurement of a condition, activity or function relating to a human or animal [1]. It has an inherent advantage of continuous patient health monitoring through wired and wireless communication. In the present situation, human wearable devices are widely used in the medical field for the purpose of health monitoring and diagnostics [2,3,4,5]. The remote monitoring module is able to track real time information of the physical condition as well as movements. Sensors are integrated into many types of radio wave devices such as capsules, textile and elastic bands, and are implanted or directly adhered to human skin in combination with external devices for wireless monitoring of heart rate [6,7,8], vital signs [9], blood pressure [10,11], body temperature [12], electrocardiograms [13,14,15,16,17] and so on.

A general architecture of an overall remote monitoring system is presented in Figure 1. The monitoring module can transmit the measured data through Bluetooth, WLAN or Wi-Fi to a computer or a mobile device for storage and data analysis.

An antenna is one of the key components in the remote monitoring system. For example, implantable radio wave devices must contain implantable antennas in order to allow in-body communication. However, the implantable antennas face more difficulties and challenges than a wearable antenna for on-body communication due to the complex human tissue conditions. The human body is often referred to as a “lossy” medium due to the performance of antennas that is easily affected by the conductivity (*σ*) and permittivity (ε*_r_*) inside the human body [18,19,20,21]. Due to the lossy characteristic of the human body, the radio wave degradation has been very severe. Hence, radio wave devices need to be improved.

One example of the developed radio sensors for health monitoring systems is the wireless capsule endoscopy (WCE) that is equipped with sensing elements and an image capture system. The sensing element consists of a very small antenna that acts as a transmitter and receiver for data transfer from human body. For such a device, it is very critical for the antenna to be very small in size and highly efficient. 

Previously, a number of antenna designs have been developed and tested in human phantoms for WCE applications. Some of the options for implantable radio sensor antennas that can be installed in WCE include embedded [22,23,24,25], conformal [26,27,28,29], a planar inverted-F (PIFA) antenna [30], a meander antenna [31], a patch antenna [32] and a spiral antenna [33]. However, a normal-mode helical antenna (NMHA) is found to be a promising candidate due to its high efficiency and small size. Thus, it can be a potential candidate to be implanted inside the human body. The targeted application of NMHA is in the WCE, and the antenna is expected to operate inside the human stomach, fat and skin tissues.

Many researches have been carried out to study the electrical characteristics and performance of NMHAs inside a human body. The self-resonant structures, the effects of material constants to antenna diameters, electric field, magnetic field, input resistance (*R_in_*) and efficiency, and the relations of antenna setting conditions to input impedance and radiation patterns have also been clarified through electromagnetic simulations [34]. However, the effects of *σ* and *ε_r_* as the key factors in the human body on the antenna input resistance are yet to be determined. 

In previous studies, it was already clarified through simulation that when the conductivity of dielectric material was increased, the input resistance also increased [19]. Another paper also presented that the *σ* and *ε_r_* of human tissue are among the factors that have significant effects to the surface impedance, which has further affected the *R_in_* of NMHAs in human tissue [35]. However, the exact relation between them is still unknown. In order to investigate the reason of increase in input resistance, the current distribution data are obtained in this paper and the inverse relation of current amplitude and input resistance is found out. As an influence of the increment of input resistance, the increase in bandwidth is also shown.

In this paper, a systematic study on the correlation of input resistance with other parameters is presented. The dependency of *R_in_* to *σ* and *ε_r_* are comprehensively discussed. Additionally, the effects of *R_in_* to bandwidth (BW), electrical field and current distributions are numerically clarified. Furthermore, the measurement of NMHA inside a human body phantom at various body parts with different permittivity and conductivity is presented for analysis and validation. A helical antenna and two human stomach phantoms were fabricated for measurement purposes.

## 2. Fundamental Equations of NMHA

The NMHA structure and electrical current model are shown in Figure 2. The antenna structural parameters such as the height, diameter and number of turns are indicated by H, D and N, respectively. The spiral current produced by the antenna can be divided into an electric current source from the small dipole, and a magnetic current source of the small loop.

The small dipole and small loop produce input impedance of *R_rD_* − *jX_D_* and *R_rL_* + *jX*_L_, respectively. The input impedance of the antenna (*Z_ant_*) is expressed by Equation (1):(1)$$Z_{ant}=R_{rD}+R_{rL}+R_{rl}+j(X_{L}-X_{D})$$

The terms *R_rD_* and *R_rL_* represent the radiation resistance component of the antenna; they correspond to the resistance of the small dipole and resistance from the small loops, respectively. The dissipated resistance due to ohmic resistance of the antenna wire is denoted by *R_l_*.

NMHA is self-resonant when the inductive reactance from the dipole source, *X_L_*, is equal to the capacitive reactance from the loop source, *X_D_*, as given in Equation (2):(2)$$X_{L}=X_{D}$$

Prior study has established the self-resonant equation in human body conditions [36], as in Equation (3) where the left and right side terms correspond to the *X_L_* and *X_D_*, respectively.
(3)$$\sqrt{\frac{\mu_{r}}{\varepsilon_{r}}}\frac{600\pi \times 19.7N{\left(\frac{D}{\lambda_{g}}\right)}^{2}}{9\frac{D}{\lambda_{g}}+20\frac{H}{\lambda_{g}}}=\sqrt{\varepsilon_{r}\mu_{r}}\frac{125\frac{H}{\lambda_{g}}}{\pi N{\left(1.1\frac{H}{\lambda_{g}}+\frac{D}{\lambda_{g}}\right)}^{2}}$$

The wavelength inside the material, *λ_g_*, is calculated as Equation (4). The equation is dependent on the wavelength in free space, *λ_o_*:(4)$$\lambda_{g}=\frac{\lambda_{o}}{\sqrt{\varepsilon_{r}}\mu_{o}}$$
where *ε_r_* is the dielectric permittivity of human body part and *µ_o_* is permeability in free space.

In a self-resonant situation, the antenna impedance becomes a pure resistance as shown in the next expression: (5)$$R_{ant}=R_{rD}+R_{rL}+R_{l}$$

In the case of in-body conditions, the input resistance of the body *R_in(body)_* can be expressed as shown in Equation (6):(6)$$R_{in(body)}=R_{ant}+R_{loss(body)}$$
where *R_loss(body)_* represents the additional resistance produced by the conductivity (*σ*) of a human body.

## 3. Electromagnetic Simulation of NMHA in Human Body Conditions

### 3.1. Simulation Model

The dielectric environment is modeled using the electromagnetic simulator FEKO 2018. The NMHA is embedded inside the center of a homogenous cylindrical human body phantom as illustrated in Figure 3. The antenna is covered with a cylindrical capsule to ensure non-direct contact with the dielectric material. An air gap is set to be 1mm between the antenna and the body phantom.

### 3.2. Simulation Parameters

The simulation parameters are summarized in Table 1. The operating frequency of 402 MHz is selected based on the medical implant communication service (MICS). Since this study is related to biomedical applications, the implanted antennas are assumed to operate in the frequency band of 402–405 MHz, which is recommended by the European Radiocommunications Committee (ERC) for ultra-low-power active medical implants [37].

The permittivity values of *ε_r_*_1_ = 67.5, *ε_r_*_2_ = 11.6 and *ε_r_*_3_ = 46.7 [38] correspond to a human stomach, fat and skin, respectively. The conductivity of the human body is simulated from *σ* = 0 to *σ* = 1.1 in order to study the effect on the input resistance.

## 4. Simulation Results

The NMHA configuration as shown in Figure 3, with the selected parameters as listed in Table 1, was simulated. The antenna parameters such as the self-resonant structure, input resistance, current distribution, voltage standing wave ratio (VSWR) and bandwidth were analyzed from the simulation results.

### 4.1. Self-Resonant Structure

Figure 4 shows the self-resonant structures of the NMHA in human body equivalent mediums, which are stomach, fat and skin with *ε_r_*_1_ = 67.5, *ε_r_*_2_ = 11.6 and *ε_r_*_3_ = 46.7, respectively. At this point, the *Z_in_* is pure resistance and the diameter, *D*, can be obtained by fixing the value of the height, *H*, and the number of the turn, *N*. The *D* and *H* will be analyzed in terms of *D*/*λ_g_* and *H*/*λ_g_*, which are normalized to wavelength in the human body (*λ_g_*).

Based on the figure, there is no significant difference in the resonant curve for different values of *σ*. Thus, it can be considered that conductivity does not change the dimensions of NMHA, in terms of *D* and *H*. Therefore, the conductivity of the human body tissues is considered to have no effects on the resonant condition of NMHA. For further analysis, the NMHA structure at point D is selected, which represents the NMHA structure in human stomach tissue (*ε_r_*_1_ = 67.5) for *σ* = 1.1 S/m that is suitable for WCE application. At this point, *λ_g_* = 90.77 mm, *H*/*λ_g_* = 0.2 mm and *D*/*λ_g_* = 0.1209 mm.

### 4.2. Input Resistance

In this section, the antenna input impedance is calculated. Figure 5 shows the Smith chart display of the input impedance at the resonant point. At this point, the input impedance becomes pure resistance (*R_in_*) at the resonant frequency, *f*_o_ = 402 MHz. The solid black line and dashed line indicate the values of input resistance at *σ* = 0 and *σ* = 1.1, respectively. It is observed that the input resistance is increased from *σ* = 0 (*R_in_* = 0.8 Ω) to *σ* = 1.1 (*R_in_* = 29 Ω). To see the trends more clearly, Figure 6 is plotted, elaborating data of *R_in_* at more discrete points of *σ*.

Based on Figure 6, the antenna input resistances at *σ* = 0 are *R_in_*_(_*ε_r1)_* = 0.8 Ω, R_in(_*ε_r2)_* = 1.1 Ω and R_in(_*ε_r3)_* = 0.8 Ω. These values are very small and almost equal to 0 Ω. With reference to Equation (5), this explains the input impedance behavior that is almost equal to the input resistance of the body *R_in(body)_* with very minimal existence of antenna input resistance. At the same time, at higher *σ*, a significant increase in the *R_in_* is recorded. For stomach and skin, the *R_in_* experiences a steady rise as *σ* increases. A different tendency is shown in the lower permittivity condition (fat, *ε**_r_* = 11.6), where the *R_in_* rapidly increases from *σ* = 0 to *σ* = 0.3 and then stays constant at *σ* = 1.1. The input resistance in the human stomach with a dielectric permittivity of *ε**_r_* = 67.5 will be explained specifically in this paper at points A, B, C and D, which represent the conductivity of *σ* = 0, *σ* = 0.3, *σ* = 0.6 and *σ* = 1.1, respectively. From Figure 6, the input resistance at points A, B, C and D is *R_in(σ=0)_* = 0.8 Ω, *R_in(σ=0.3)_* = 11.8 Ω, *R_in(σ=0.6)_* = 20.5 Ω and *R_in(σ=1.1)_* = 29 Ω, respectively.

As a summary for the dependency of *R_in_* on *σ*, it is clear that the input resistance has increased with the increase of the *σ* value. For the dependency of *ε_r_* on the input resistance, further study of other data such as electric field distributions are needed.

### 4.3. Electric Field

In order to investigate the changes in *R_in_*, the effects of electric field distribution shall be examined. The changes in the near field distributions are shown in Figure 7a–c. When the conductivity is increased, the electric field distribution areas and electric field strengths are decreased. However, this information is not sufficient to explain the increment of *R_in_*. Hence, the current distributions on the antenna should be investigated.

### 4.4. Current Distribution

The current distribution along the wire is analyzed as shown in Figure 8. The Maximum current (*I_max_*) is observed at the center of the wire (red color), approximately at H/2. The current distribution is tapered towards the end of the antenna, where *I* = 0 A (dark blue color). A strong current area is concentrated at the center of the antenna for *ε**_r_*_1_ = 67.5, *σ* = 0 as compared to *σ* = 0.3, *σ* = 0.6 and *σ* = 1.1. The blue color along the wire in *σ* = 0.3, *σ* = 0.6 and *σ* = 1.1 shows that a very low current is distributed at higher *σ*.

For further analysis, the *I_max_* for *ε**_r_*_1_ = 67.5 with different values of *σ* is plotted in Figure 9. 

As seen in the graph, the maximum current at point A, B, C and D is *I_max(σ=0)_* = 1.12 A, *I_max(σ=0.3)_* = 0.08 A, *I_max(σ=0.6)_* = 0.05 A, and *I_max(σ=1.1)_* = 0.03 A, respectively. It is obviously shown that the *I_max_* is reduced as the value of *σ* increases. 

Because *I_max_* changes with respect to the change of *σ*, *R_in_* can be calculated by Equation (7). It is noted that the input voltage, *V* was set to be 1 V:(7)$$R_{in}=\frac{V}{I_{\max}}$$

The calculated *R_in_* by *I_max_* results are plotted in Figure 10. From the graph, it is clearly shown that the calculated *R_in_* agree well with the simulated *R_in_* for *ε**_r_*_1_ = 67.5, as illustrated previously in Figure 6. The good agreement of Equation (7) and the results of *R_in_* in Figure 6 demonstrates the reason for the increase of *R_in_*. It can be summarized that *I_max_* is inversely proportional to *R_in_* as *σ* increases. However, detailed discussion on the *I_max_* reduction in lossy material will be made in future works.

### 4.5. Bandwidth

The bandwidth can be expressed by *R_in_* as follows. The *R_in_* and antenna Q factor are related by Equation (8):(8)$$Q=\frac{X_{L}}{R_{in}}$$

Here, *X_L_* is the inductive reactance as shown in Equations (2) and (3). Then, the bandwidth (BW) at the specified VSWR is expressed by Equation (9):(9)$$BW=\frac{VSWR-1}{Q\sqrt{VSWR}}$$

As a result, BW becomes proportional to *R_in_* and is dependent on conductivity as expressed by Equation (10):(10)$$R_{in(\sigma )}\propto BW_{\left(\sigma \right)}$$

The simulation results of the antenna VSWR characteristics for *ε_r_*_1_ = 67.5 are shown in Figure 11. It is shown that VSWR increases as the conductivity increases for *ε_r_*_1_ = 67.5 (VSWR_(*σ*=0)_ = 0.5 MHz to VSWR_(*σ*=1.1)_ = 21 MHz).

Figure 12 explains the relation of the simulated *R_in_* to the fractional bandwidth (FBW) by conductivity. The graph clearly shows the direct proportional characteristics of FBW to *R_in_* as in Equation (10).

### 4.6. Radiation Pattern

The radiation pattern is a graphical representation of the radiation (far-field) properties of an antenna. The simulated radiation patterns for *ε_r_*_1_ = 67.5, at point A (*σ* = 0) and point D (*σ* = 1.1) are shown in Figure 13. Antenna gain (*G_A_*) values at point A and point D are −5.357 dBi and −21.78 dBi, respectively.

When *R_a_* is antenna resistance, and *R_in(σ)_* is the input resistance in the human body, then antenna efficiency, *η_a_*, can be simplified to below expression:(11)$$\eta_{a}=\frac{R_{a}}{R_{in(\sigma )}}$$

Then, antenna gain, *G_A_*, can be expressed by the next equation:(12)$$G_{A}=\eta_{a}+\Gamma +G_{D}\;(\mathrm{dBi})$$

Here, *G_D_* is the antenna gain for a short dipole antenna and the value is 1.8 dBi, where, *Г* is the reflection of the material surface. The calculation of *η_a_* and *G_A_* values for *ε_r_*_1_ = 67.5, *σ* = 0 and *σ* = 1.1 of Figure 13 are shown in Table 2.

In Table 2, *G_A_* value for both *σ* = 0 and *σ* = 1.1 become almost identical to the *η_a_* value. The reason is perhaps that the *Г* is around 2 dB, thus *Г* and *G_D_* is cancel each other out. This explains the close values for *G_A_* and *η_a_*.

## 5. Measurement Setup

### 5.1. Antenna Fabrication

For measurement purposes, a NMHA was fabricated. A Metallic copper wire with a conductivity of 58 × 10^6^ (S/m) was used to construct the antenna. In order to fabricate the antenna, ‘point D’ was selected at *H*/*λ**_g_* = 0.2 as shown in Figure 4. Hence, the *H* and *D* of the antenna were determined and the fabricated antennas are shown in Figure 14. 

Because the NMHA size is very small, antenna performance is affected by the leakage current on the coaxial cable. In order to suppress the leakage current on the outside of the outer conductor, NMHAs are soldered to a *balun*. By attaching the *balun*, leakage current is sufficiently reduced and NMHA performance is correctly achieved. In this mode, the currents on the inner conductor and on the inside of the outer conductor are equal in magnitude and opposite in direction.

### 5.2. Phantom Fabrication and Measurement

In this paper, two human body phantoms that are equivalent to biological human stomach properties were fabricated. Each of the phantoms consists of two parts which represent the top and bottom part. The target was to fabricate two human stomach phantoms with different conductivity, which represent higher (*σ* = 1.1) and lower (*σ* = 0.6) conductivity characteristics. 

The details of the chemical composition used to fabricate the human body phantoms are listed in Table 3. In order to fabricate the phantom at the desired permittivity and conductivity, a suitable quantity of chemicals is needed.

The fabricated phantoms are shown in Figure 15a,b whilst the measurement setup for the phantom dielectric permittivity and conductivity is shown in Figure 15c. From observation, the physical structure of Phantom 1 is solid and it is expected to have full stability for antenna measurement. On the other hand, for Phantom 2, the structure is a little bit sticky and watery due to the lesser amount of salt used in the fabrication.

For phantom measurement, 5 points are set on the surface of the phantoms in order to measure the dielectric permittivity and conductivity, as shown in Figure 15c. The phantom was measured using a dielectric probe. The measurement data consist of the real part of the data and the imaginary part as in Equation (13), which were then converted as in Equation (14) and Equation (15).

Here, *ε_c_* is given by Equation (13):(13)$$\varepsilon_{c}=\varepsilon_{r}+j\varepsilon_{i}$$

Moreover, *ε_i_* is related to *σ* by the next expression:(14)$$\varepsilon_{i}=\frac{\sigma }{\omega \varepsilon_{o}}$$

Hence, *σ* can be calculated by the next equation by substituting ω=2πf and $\varepsilon_{o}=\frac{10^{-9}}{36\pi }$ in Equation (14):(15)$$\sigma =\frac{\varepsilon_{i}}{45}$$

By using the average values of the real part *ε_r_,* the permittivity values for the fabricated phantoms are determined and the conductivity value can be calculated from Equation (15). The measured permittivity and conductivity values for the fabricated phantoms are Phantom 1 (*ε_r_* = 63.5, *σ* = 1.04 S/m) and Phantom 2 (*ε_r_* = 64.5, *σ* = 0.59 S/m), respectively. The graph of phantom measurement data is shown in Figure 16.

### 5.3. Antenna Measurement Setup

The antenna placement in a human body phantom is shown in Figure 17a, and the actual antenna measurement setup is shown in Figure 17b. Since an air gap must be ensured as no direct contact between the antenna and the human phantom is allowed, the antenna and the coaxial cable (*balun*) were carefully covered with plastic before placing the antenna in the middle of the human phantom. The metal isolation is important in order to ensure the accuracy of the measured results. Furthermore, as a small antenna is highly sensitive, in order to ensure the accuracy of the measurement data, the coaxial cable and antenna positions in the phantom were carefully adjusted during the measurement.

## 6. Measurement Results

### 6.1. Input Resistance

To confirm the simulated results for the input resistance at points C and D, measurements for antenna input resistance are needed. The fabricated NMHA was measured on Phantom 1 and Phantom 2. The simulated and measured antenna input resistance results for the fabricated NMHA in both Phantom 1 and Phantom 2 are shown in Figure 18.

Based on Figure 18, for *R_in_* in Phantom 1, the simulated and measured *R_in_* at point C are 20.62 Ω and 17.52 Ω, respectively, and the simulated and measured *R_in_* for Phantom 2 at point D are and 29.02 Ω and 28.57 Ω, respectively. There is a slight difference of about 3 Ω between the simulated and measured *R_in_* at point C. One of the reasons was due to the phantom’s physical structure, which was observed to be sticky and watery. This might affect the stability of the antenna setup in the phantom, because the antenna was expected to be kept still in the phantom during the antenna measurement. Thus, a slight movement of the antenna might cause the input resistance result to be shifted. Overall, it can be concluded that the simulated and measured results agreed rather well.

### 6.2. Bandwidth

The measured and simulation results of the antenna VSWR characteristics are shown in Figure 19. The simulated and measured results conform well to each other. At VSWR = 2, the simulated and measured bandwidths of the NMHA at point C (in phantom 2) are 15 MHz and 13 MHz, respectively, and correspond to a fractional bandwidth of approximately 3.73% and 3.23%, respectively. While at point D, *σ* = 1.1, the simulated and measured bandwidths are 21 MHz and 18 MHz with a FBW of 5.22% and 4.48%, respectively.

## 7. Discussion

The increase of input resistance (*R_in_*) for the normal-mode helical antenna (NMHA) in human body conditions has been examined by electromagnetic simulations and experiments. The reason of the increase of *R_in_* was found to be due to the decrease of current on the NMHA. As an effect of the rise in *R_in_* values, the increment of bandwidth was also validated. The important data in this paper are listed as follows:

By electromagnetic simulations:The relation between the increase of input resistance (*R_in_*) and dielectric permittivity (*ε_r_*) and conductivity (*σ*).The relation between input resistance (*R_in_*) and current distributions.The relation between input resistance (*R_in_*) and bandwidth (BW).

By experimental data with a fabricated body phantom:Input resistance (*R_in_*).Bandwidth (BW).

For a future research direction, *R_in_* measurements for different permittivities and conductivities such as *ε_r_* = 11.6 and *σ* = 0.3 will be conducted. Moreover, the reason of the current decreasing by the increase of conductivity will be validated through an analytical approach for electromagnetic field and current relations. 

## Figures and Tables

**Figure 1 sensors-20-00958-f001:**
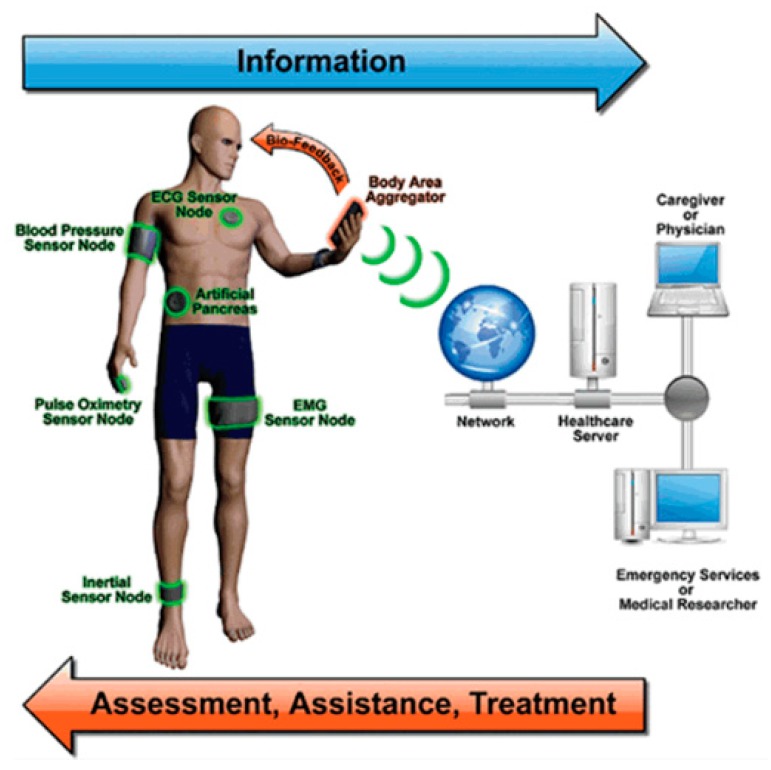
Integration of various sensors on a patient for remote health monitoring.

**Figure 2 sensors-20-00958-f002:**
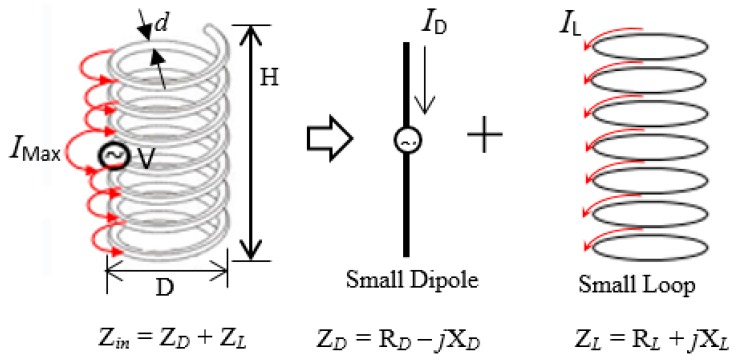
Normal-mode helical antenna (NMHA) structure and electrical current model.

**Figure 3 sensors-20-00958-f003:**
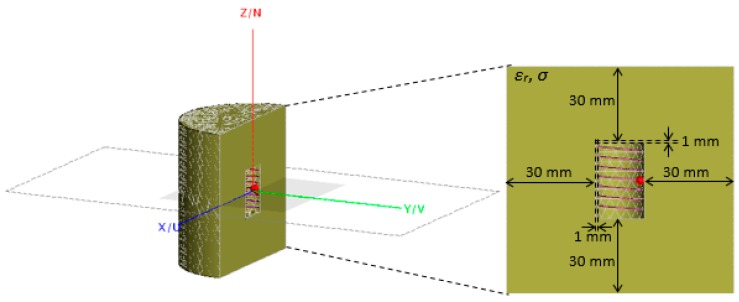
NMHA simulation model. ([2018] IEEE. Reprinted, with permission, from [18]).

**Figure 4 sensors-20-00958-f004:**
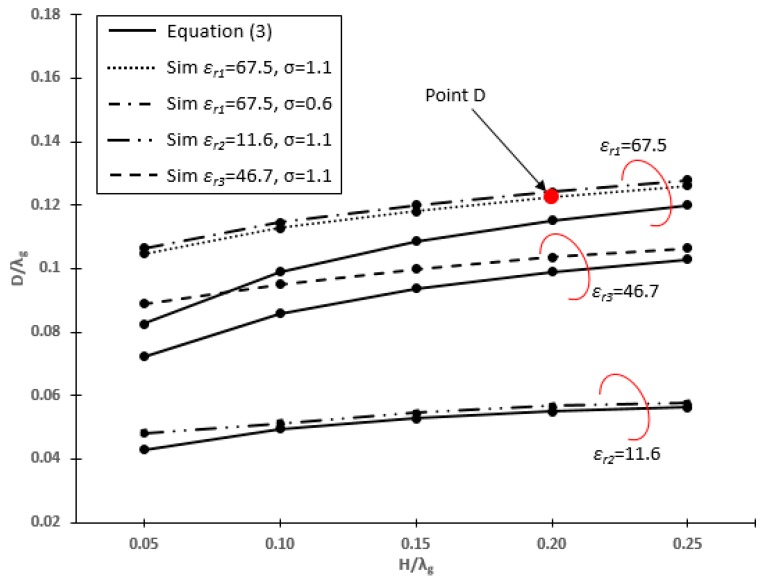
NMHA self-resonant structure.

**Figure 5 sensors-20-00958-f005:**
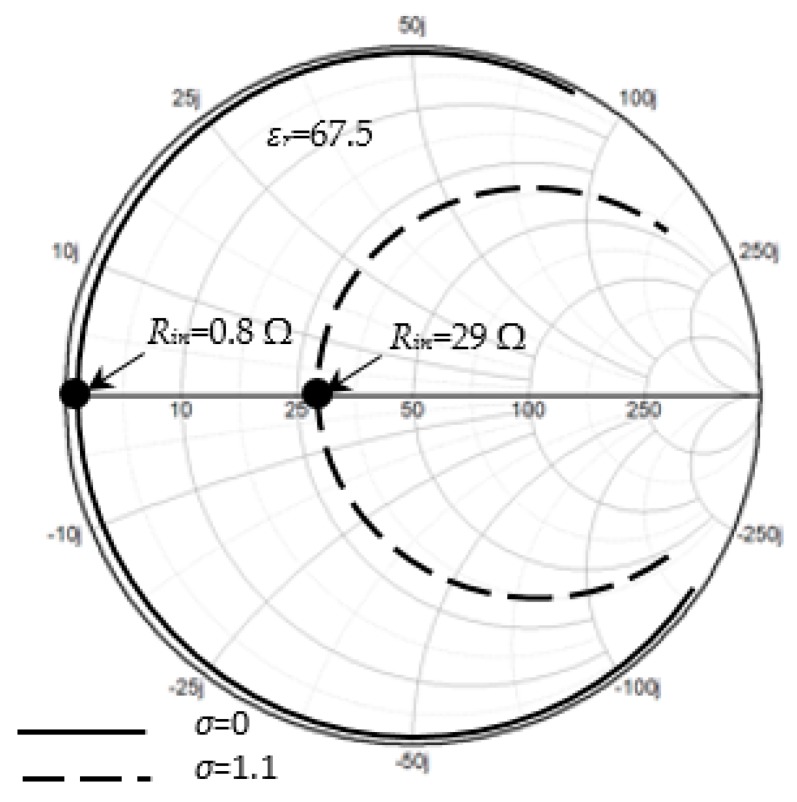
Input impedance in a Smith chart plot for ε*_r_*_1_ = 67.5 and *σ* = 0, 1.1.

**Figure 6 sensors-20-00958-f006:**
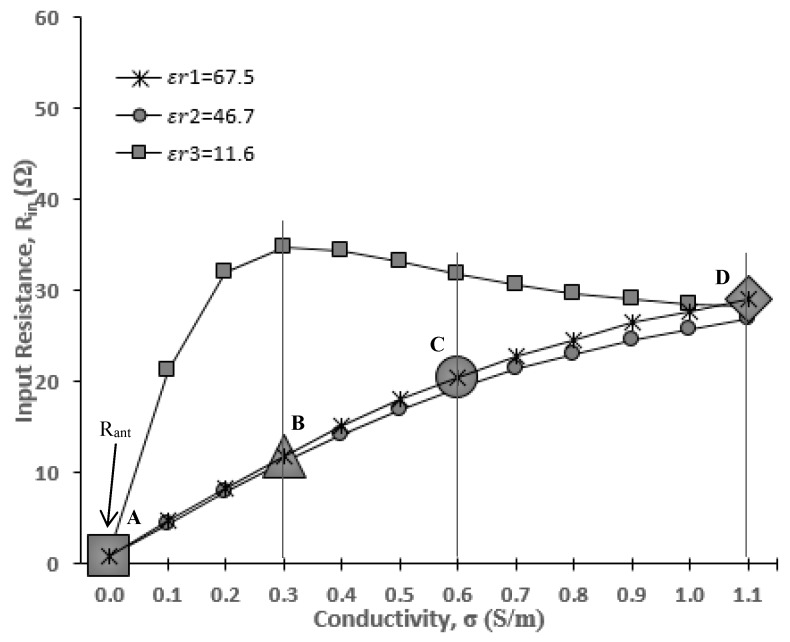
NMHA input resistance with respect to conductivity.

**Figure 7 sensors-20-00958-f007:**
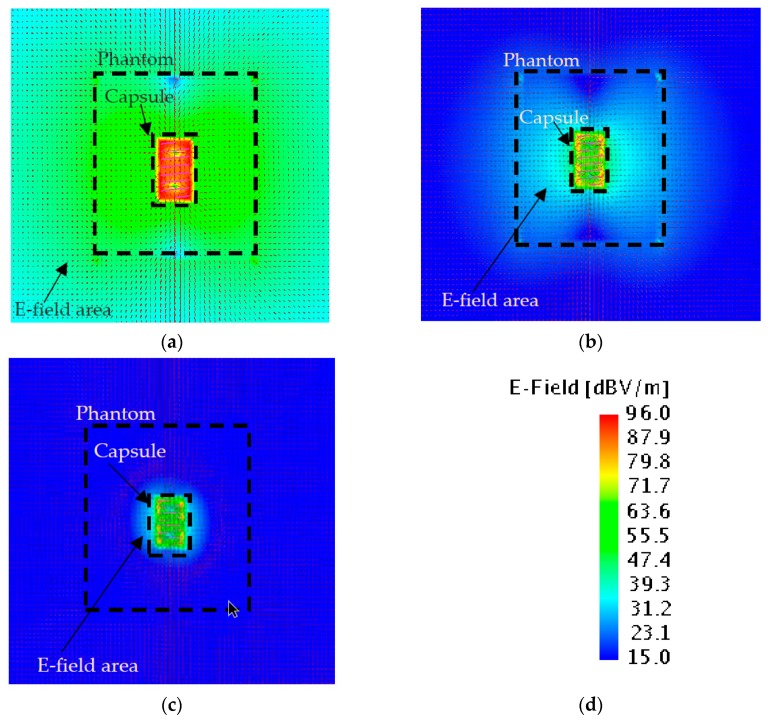
Electric field distributions of NMHA in the human stomach, *ε_r_*_1_ = 67.5. (**a**) *σ* = 0. (**b**) *σ* = 0.3. (**c**) *σ* = 1.1. (**d**) E-field legend scale.

**Figure 8 sensors-20-00958-f008:**
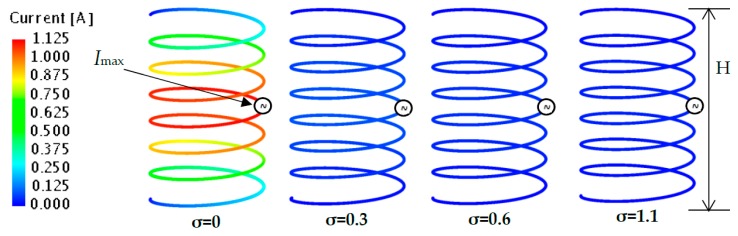
Current distribution of a NMHA wire for *ε_r_*_1_ = 67.5, (*σ* = 0, 0.3, 0.6, 1.1).

**Figure 9 sensors-20-00958-f009:**
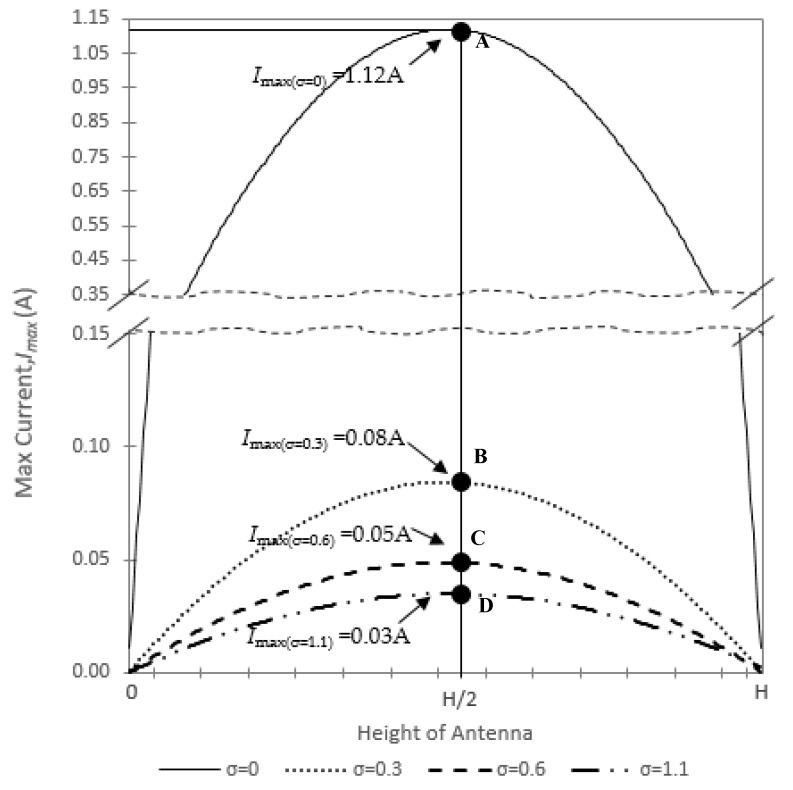
Current distribution for *ε_r_*_1_ = 67.5, (*σ* = 0, 0.3, 0.6, 1.1).

**Figure 10 sensors-20-00958-f010:**
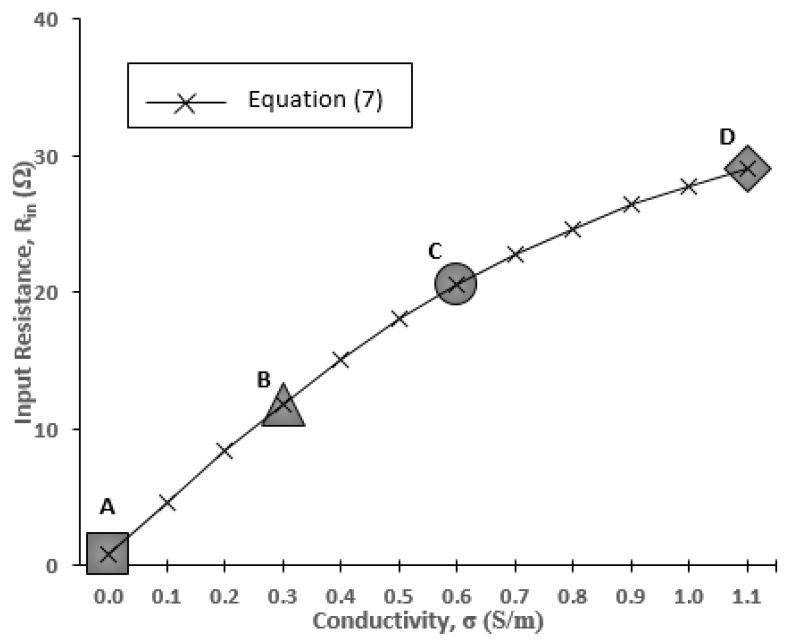
Calculated input resistance for *ε_r_*_1_ = 67.5.

**Figure 11 sensors-20-00958-f011:**
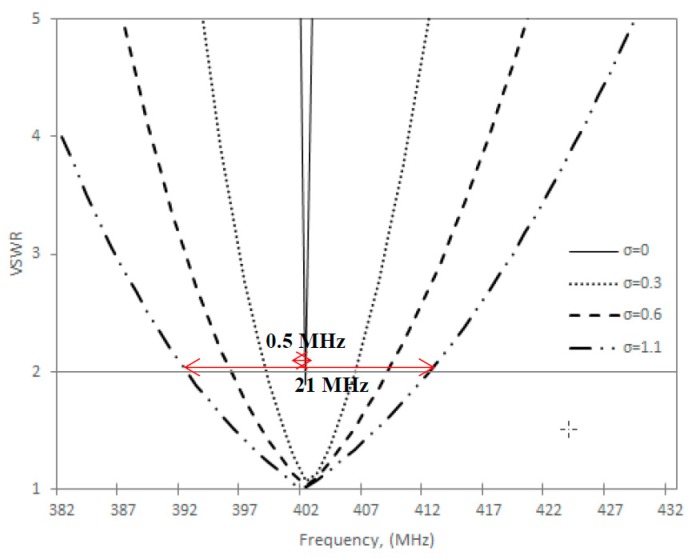
Antenna bandwidth for *ε_r_*_1_ = 67.5, *σ* = 0, 0.3, 0.6, 1.1.

**Figure 12 sensors-20-00958-f012:**
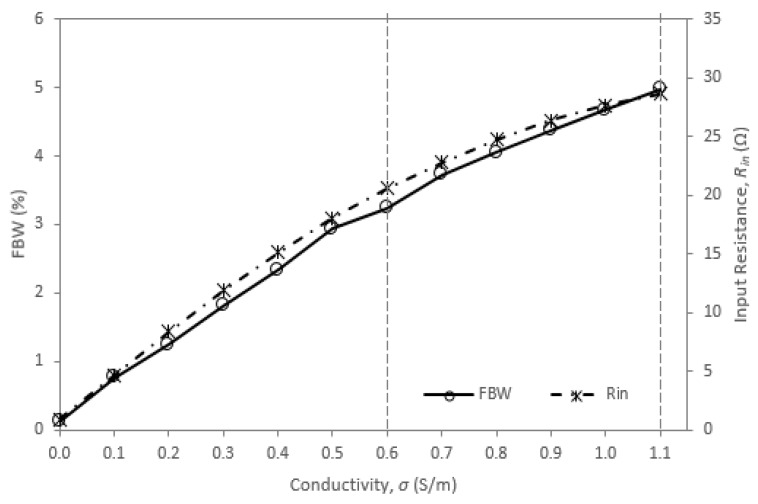
Relation of simulated *R_in_* and fractional bandwidth (FBW) by conductivity for *ε_r_*_1_ = 67.5.

**Figure 13 sensors-20-00958-f013:**
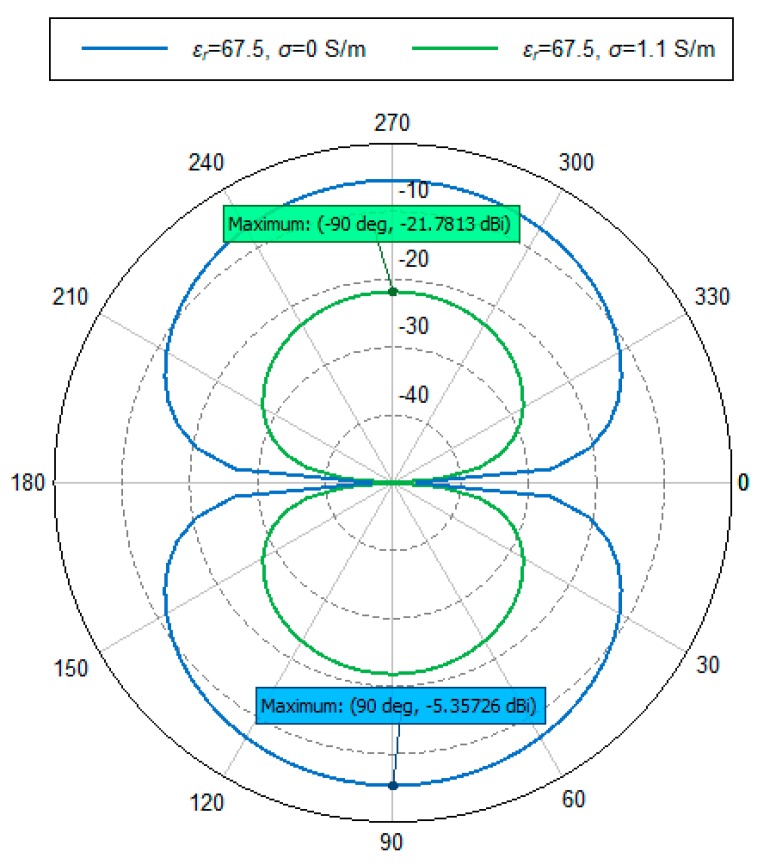
Simulated radiation pattern for *ε_r_*_1_ = 67.5, *σ* = 0, *σ* = 1.1.

**Figure 14 sensors-20-00958-f014:**
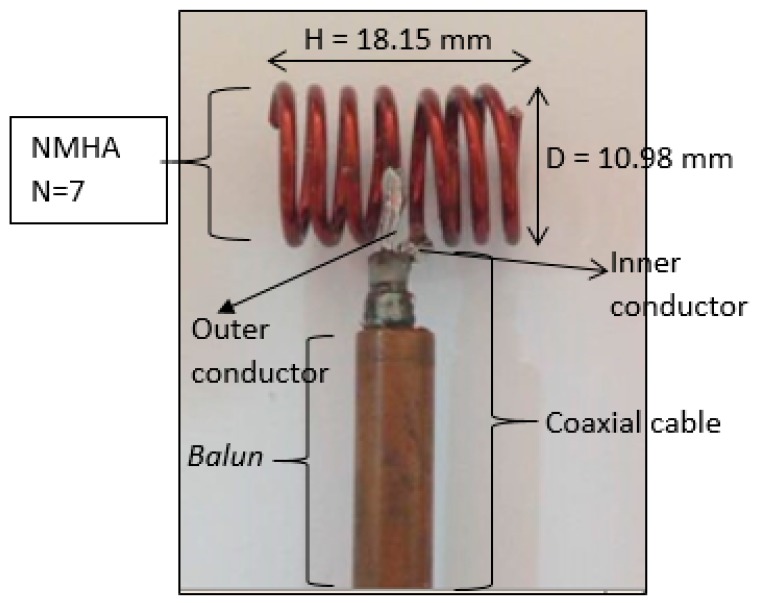
Fabricated NMHA.

**Figure 15 sensors-20-00958-f015:**
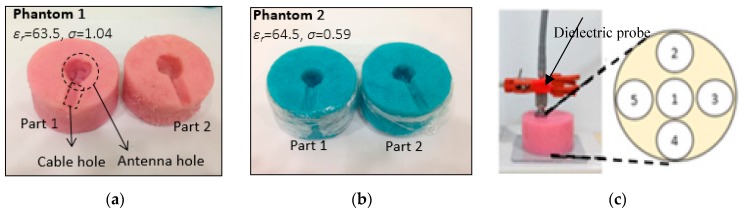
Fabricated phantoms: (**a**) human stomach phantom for higher *σ*, (**b**) human stomach phantom for lower *σ*, (**c**) phantom measurement setup.

**Figure 16 sensors-20-00958-f016:**
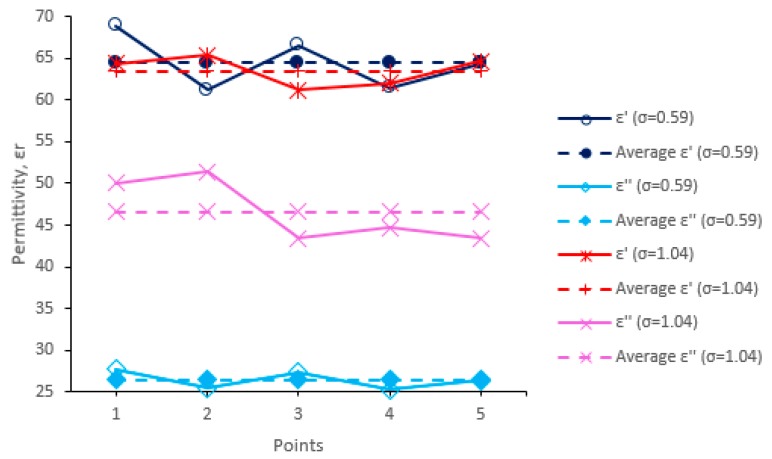
Phantom measurement data for human stomach phantoms of higher and lower conductivity.

**Figure 17 sensors-20-00958-f017:**
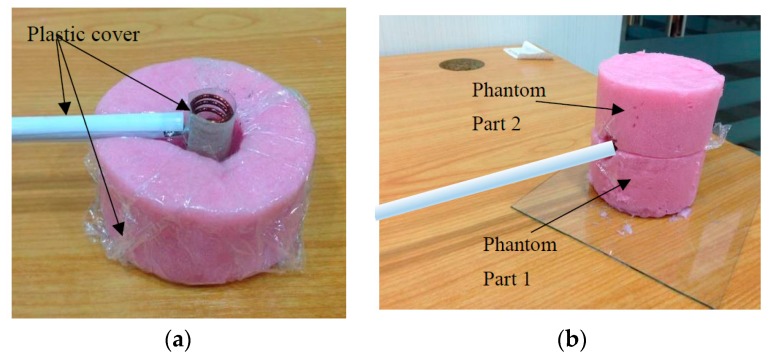
Antenna measurement setup. (**a**) Antenna placement in a phantom; (**b**) actual measurement setup. ([2018] IEEE. Reprinted, with permission, from [18])

**Figure 18 sensors-20-00958-f018:**
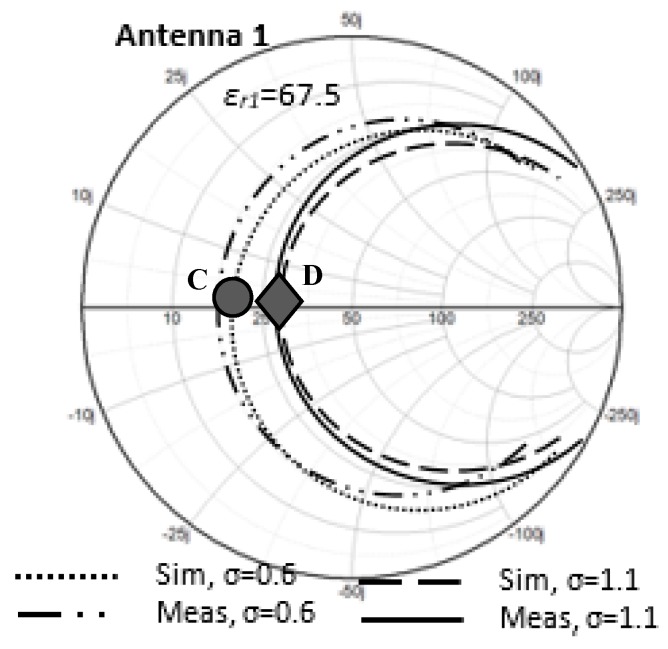
Simulated and measured input resistance for Antenna 1 in a higher conductivity (*σ* = 1.1) and lower conductivity (*σ* = 0.6) human stomach phantom.

**Figure 19 sensors-20-00958-f019:**
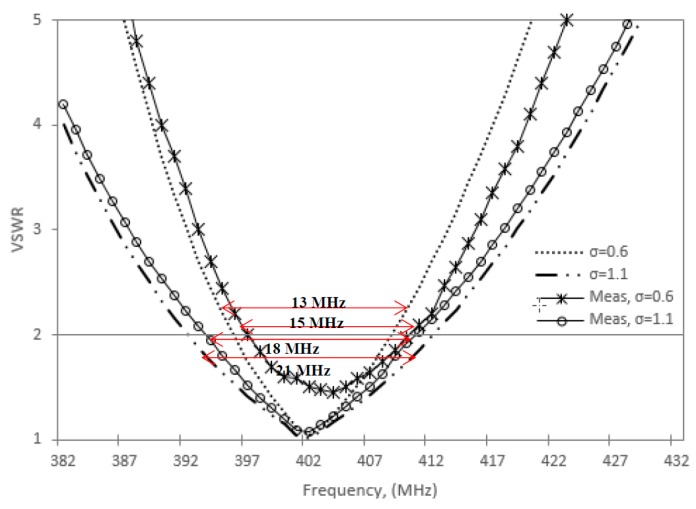
Antenna bandwidth for *ε_r_*_1_ = 67.5, *σ* = 0.6, 1.1.

**Table 1 sensors-20-00958-t001:** Simulation parameters. MICS: medical implant communication service.

Aspect	Parameter	
Simulator	FEKO 2018	Method of Moment
Frequency	MICS Band	402 MHz
Antenna	Height, H/*λ_g_*	0.2
	Diameter, D/*λ_g_*	0.0965~0.1210
	Number of turns, *N*	7
	Diameter of wire, *d*	1.2 mm
	Mesh size, Δ*l*	*λ_g_*/100
	Metallic Wire, Copper	5.8 × 10^6^ (S/m)
Dielectric (Human Body)	Permittivity, *ε_r_*	67.5, 46.7, 11.6
	Permeability, *μ_r_*	1
	Conductivity, *σ*	0~1.1
	Wavelength material, *λ_g_*	90.77 mm,
	Mesh size, Δ*m*	*λ_g_*/100

**Table 2 sensors-20-00958-t002:** Calculation of efficiency, *η_a_*.

*ε_r_*_1_ = 67.5	*R_a_* [Ω]	*R_in(σ)_* [Ω]	*η_a_* [dBi]	*G_A_* (sim) [dBi]
*σ* = 0 [S/m]	0.227	0.8	−5.47	−5.357
*σ* = 1.1 [S/m]	0.217	29.02	−21.26	−21.78

**Table 3 sensors-20-00958-t003:** Chemical composition for human body phantom fabrication.

		Human Stomach Phantom	
Materials	Function	Phantom 1 *ε_r_*_1_ = 67.5, *σ* = 1.1	Phantom 2 *ε_r_*_1_ = 67.5, *σ* = 0.6
Distilled water	Main material	400 mL	400 mL
Polyethylene powder	Permittivity	6 g	6 g
Agar	Forming material	17.5 g	17.5 g
Sodium Chloride (NaCl)	Conductivity	1.925 g	0.9 g
Xanthan	Thickener	15 g	15 g
Sodium Dehydro-acetat	Preservative	0.25 g	0.25 g

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
