# Peer review of "Increase of Input Resistance of a Normal-Mode Helical Antenna (NMHA) in Human Body Application"

_sensors, 2020, doi:10.3390/s20040958_

Round 1

Reviewer 1 Report

This work explains relationship between conductivity of human body to input resistance, current distribution and bandwidth of Normal-mode helical antenna (NMHA). The paper is written well and my opinions are: 

In line 156, Fig. 6., the simulated input resistances are changed with different values of conductivity of stomach, skin and fat, while in line 264, Fig. 16., the measured input resistances are changed with only two conductivity values of stomach, which cannot verify the simulated results. The situation of bandwidth analysis has the similar problem. Therefore, more kinds of human body phantom with different conductivity values should be considered. As we know, a biocompatible layer should be added to avoid direct contact between the NMHA and the human body tissues such as stomach, skin and fat. More importantly, biocompatible layer isolates the proposed antenna from the stomach, which means variation of conductivity values may not influence both input resistance and bandwidth of the NMHA. Therefore, if the proposed NMHA is covered with a biocompatible layer, are the results still be right? Please validate.

Thanks!

Reviewer 2 Report

The manuscript presents a helical dipole. The size is compact and it raises questions regarding the accuracy of the measured results. For the claimed frequency the size is too small.

It is highly likely that the radiation efficiency and the gain to be very low something that limits the usefulness of the implantable antenna.

Return loss measurement is missing

Radiation pattern measurement is missing

Gain/efficiency is missing

It must be explained why the specific phantoms (out of many) were chosen.

The manuscript must be revised and resubmitted to be complete. Even if it was complete the novelty would   still be at question.

Reviewer 3 Report

This paper tries to explain the reason why the input resistance of a Normal Mode Helical Antenna(NMHA) would increase when the antenna is placed inside a human body by some simulated results and experiments. I recommend that this manuscript should be revised in following aspects. 

The literature review is not very intensive. Introduction section should include more references about what is the current situation in this field.

The theory behind the phenomenon is weak. Another article paper of the authors, titled “Effects of the permittivity and conductivity of human body for normal-mode helical antenna performance”, gave the electromagnetic simulations. This paper includes the experimental results. But the authors did not explain what is the reason behind the phenomenon, and did not establish a theoretic model to describe the mechanism.

Figure 1, 2 and 3 should be replaced by figures with better resolution. Please use some sharp figures in this paper.

This paper studied the input resistance of a NMHA influenced by placing an electric conductor in the near filed of the antenna. It is similar to previous study of a dielectric-loading normal mode helical antenna. I recommend that this paper should provide some comparisons between the dielectric-load helical antenna and the NMHA.

According to Section4.4, the increase of input resistance is mainly due to the decrease of the distributed current. This phenomenon is only verified by simulation, and authors do not provide more insights in the function between the current distribution and the conductivity change of the surrounding dielectric load qualitatively or quantitatively. The authors could use electromagnetic theories to explain the phenomenon for strengthening the theory part of this paper, or it would be so much similar to the previous work of the authors.

Line 67. What’s the difference between “spiral antenna” and “helical antenna”?

Line 67. The sentences are confusing, please revise them.

Line 146, Figure 4. Three wires with label “Equ” could not be distinguished. Please label each clearly.

Round 2

Reviewer 1 Report

Thanks

Reviewer 2 Report

The manuscript can be accepted in this revised form.

Reviewer 3 Report

Accept in present form